# Current Status of Robot-Assisted Revisional Bariatric Surgery

**DOI:** 10.3390/jcm11071820

**Published:** 2022-03-25

**Authors:** Carolina Vanetta, Nicolás H. Dreifuss, Francisco Schlottmann, Alberto Mangano, Antonio Cubisino, Valentina Valle, Carolina Baz, Francesco M. Bianco, Chandra Hassan, Antonio Gangemi, Mario A. Masrur

**Affiliations:** Division of General, Minimally Invasive and Robotic Surgery, Department of Surgery, University of Illinois at Chicago, Chicago, IL 60612, USA; dreifuss@uic.edu (N.H.D.); fschlo2@uic.edu (F.S.); amangano@uic.edu (A.M.); cubisino@uic.edu (A.C.); vvalle5@uic.edu (V.V.); cbaz@uic.edu (C.B.); biancofm@uic.edu (F.M.B.); chandrar@uic.edu (C.H.); agangemi@uic.edu (A.G.); mmasrur@uic.edu (M.A.M.)

**Keywords:** revisional bariatric surgery, minimally invasive bariatric surgery, robot-assisted bariatric surgery, laparoscopic revisional bariatric surgery, robot-assisted revisional bariatric surgery

## Abstract

Bariatric surgery has been demonstrated to be effective in achieving significant weight loss and remission of obesity-related comorbidities. However, a percentage of patients fail to lose enough weight, regain weight, or experience postoperative complications, requiring additional interventions. Revisional bariatric surgeries (RBS) involve the wide spectrum of procedures that aim to treat complications of the index operation or achieve further weight loss. These are technically challenging procedures due to adhesions of the internal organs, reduced working space, and a distorted anatomy. Indications, timing, and type of operation for RBS are not standardized, and there is no consensus on the best surgical approach. Some authors claim a robotic platform could be advantageous in these types of procedures that are performed in reduced, deep operating fields, or those requiring precision and accuracy. This review examines the most current and representative literature on the outcomes of robot-assisted RBS. Included studies demonstrate the safety and feasibility of the robotic approach for RBS. However, long operative times and high costs remain major drawbacks of the device. Finally, if we consider that many centers have not yet completed the learning curve for robot-assisted RBS, the potential for improved outcomes seems promising.

## 1. Introduction

Obesity rates are on the rise worldwide. According to the 2021 National Health Statistics Reports from the Centers for Disease Control, the prevalence of obesity reaches 41.9% among adults and 19.7% among children and adolescents aged 2–19 years old [1]. Specifically, severe obesity, corresponding to a body mass index (BMI) of 40 or higher, is present in 9.2% of United States adults [2]. The relationship between obesity and the development of various well-known comorbidities, including cardiovascular diseases, diabetes, liver disease, chronic kidney disease, osteoarthritis, and cancer, positions this condition as one of the top preventable causes of mortality. On this matter, intense lifestyle modification and pharmacotherapy usually fail to achieve long-term sustained weight loss and remission of obesity-related comorbidities (10% weight loss at 1 year and 5.3% at 8 years) [3]. On the contrary, bariatric surgery has been demonstrated to be effective at achieving long-term weight loss (up to 77% of excess weight at 1 year and more than 50% at 10–20 years [2,4]), remission of obesity-related comorbidities, and reducing the incidence of major cardiovascular events [5]. Despite these alarming figures, the American Society for Metabolic and Bariatric Surgery estimates indicate that less than 1% of the population eligible for bariatric surgery actually received it in 2019 [2], a dissociation that is believed to correspond to unequal access to surgery, related to income and education level, among other factors [6].

Nevertheless, the rising demand for bariatric procedures has resulted in an inevitable increase in secondary (revisional) bariatric surgeries (RBS) in the last decade. Accounting for 6.0% of the bariatric procedures in 2011, they peaked at 16.7% in 2019 in the United States [7]. Although the trending increase in the percentage of RBS relative to other bariatric procedures is difficult to explain, it could be related to multiple factors such as higher rates of obesity and primary bariatric surgeries, a shift in the types of bariatric procedures being performed, failure of some procedures to achieve sustained weight loss, increase in the available literature supporting the benefits of revisional bariatric surgery, and an increased experience with revisional cases.

RBS encompasses a wide spectrum of surgical procedures performed after the failure of a primary bariatric operation. Basically, the indication for RBS falls into one of the following scenarios: inadequate weight loss, weight regain, persistence or recurrence of comorbidities, and postoperative complications of the primary bariatric procedure. The types of revisional procedures are as follows: (a) revision or correction, which implies the abdominal exploration and re-evaluation of the anatomy, usually in an attempt to address refractory symptoms; (b) conversion, in which a specific bariatric procedure is converted into another type of bariatric procedure; and (c) reversal, where the original anatomy is reestablished [8]. Hence, the indications and types of RBS vary according to the index procedure and the necessity of a subsequent intervention. Recent estimates indicate that the incidence of RBS ranges from less than 5% to 26% [9]. Revisional rates according to the primary procedure reported in the literature are as follows: 40–60% for adjustable gastric banding (AGB), 25–54% for vertical banded gastroplasty, 30% for sleeve gastrectomy (SG), 10–20% for Roux-en-Y gastric bypass (RYGB), and 5% for biliopancreatic diversion with duodenal switch [10,11,12].

RBS are technically demanding procedures that require surgical expertise and are better managed in referral centers. Index procedures leave inflammation and adhesions, which, together with the excessive intra-abdominal fat, thick abdominal wall, and a voluminous liver, reduce the working space. Additionally, the altered anatomy and the need to perform complex interventions (i.e., re-anastomosis) pose a great challenge. Once an RBS is indicated, the decision regarding the best approach follows. Should the revisional surgery be conducted open, laparoscopic, or robot-assisted? Currently, the standard of care for primary bariatric procedures is the minimally invasive approach due to its advantages, namely: reduced postoperative pain, shorter length of hospital stay (LOS), and enhanced recovery [13]. However, indication, type of procedure, and surgical approach are not clearly standardized in RBS. Regarding this matter, certain authors claim that a robot may have technical advantages in these operations in which fine dissection and laborious procedures are required. In the present review, we rigorously illustrate the outcomes of a robot in RBS with the intent to elucidate its current status.

## 2. Indications for Revisional Bariatric Surgery and Preoperative Evaluation

The indication for revisional bariatric surgery can be divided into two groups: weight and comorbidity-related (also defined as “failure” of the primary procedure), and complication-related. If we consider the primary bariatric procedure, the indication, and the options among the revisional procedures (corrections, conversions, and reversals), a wide variety of surgical alternatives can be described (Table 1).

Unfortunately, there is no consensus in the definition of “failure” of bariatric surgery. Weight-related failure typically involves either inadequate weight loss or weight regain. The systematic review by Mann et al. [14] determined that the most frequently used definition of inadequate weight loss was <50% excess weight loss (EWL) at 18 months. The second most frequent definition was <25% EWL without a specific time-frame. In addition, weight regain has been defined as progressive weight increase after the achievement of an initially successful weight loss or nadir weight [15]. However, some authors have argued that the percentage of EWL is highly dependent on the preoperative BMI and fails to reflect the real success. Instead, they propose the percentage of the total weight loss, with a lower limit of 20%, as the best indicator of success [16,17]. In any case, the majority of studies fail to report the parameters and values used to define failure [14].

As is expected, restrictive procedures have the highest revision rates, whereas malabsorptive procedures have the lowest. Accordingly, weight loss after RBS for purely restrictive procedures reaches higher percentages (up to 70% EWL) [18,19,20,21,22] than for malabsorptive procedures (around 50% EWL) [23]. The presence of comorbidities is of high importance and should prompt the indication of RBS. Brolin et al. [23] reported a remission rate of 100% after RBS, in which ≥50% EWL was achieved, and 89% in those who did not achieve this value. Although the type of RBS to be performed is not standardized, many authors recommend conversion to RYGB after a failed restrictive procedure [20,24].

As the failure of bariatric surgery is multifactorial, a thorough multidisciplinary evaluation must be conducted before embarking on a revisional operation. Preoperative evaluation intends to identify the functional and anatomical causes of the failed and/or complicated primary procedure (stricture, dilation, stenosis, ulcers, gastroesophageal reflux disease, etc.), and should include at least an upper endoscopy and a contrast upper gastrointestinal study. Other studies, such as manometry, computed tomography scan, pH monitoring, and gastric emptying studies, may also be required according to the patient’s symptoms. Moreover, it is of paramount importance to consider the psychological and behavioral status of the patient before indicating an RBS. It is well known that weight loss failure may be linked to inadequate dietary habits and lifestyle that, if left unattended, would result in repeated failure [12].

## 3. Revisional Bariatric Surgery: A Challenging Operation

Revisional surgery implies operating on the same structure for a second time. Hence, increased perioperative complications should be expected. On this matter, Howell et al. [25] reported significantly higher morbidity rates for revisional vs. primary bariatric surgery (14.8% vs. 3.9%). Deylgat et al. [26] compared 652 patients undergoing RYGB as a primary procedure with 72 patients undergoing revisional RYGB, and found a similar morbidity and LOS. However, intraoperative complications such as serosal tearing, important bleeding, and suture dehiscence were significantly higher in revisional cases (11.11% vs. 3.22%). Zhang et al. compared 172 patients who underwent revisional RYGB with 172 paired primary RYGB patients. Higher estimated blood loss (463.7 mL vs. 113.3 mL), longer operative time (272.5 vs. 175.5 min), and more than doubled LOS (5.6 vs. 2.5 days) were found during revisional cases [27]. In another retrospective, single-center study, increased postoperative complications (41% vs. 15%), reoperations (10.8% vs. 5.4%), and prolonged LOS (4 vs. 2 days) were found after revisional surgery [28]. Nevertheless, revisional patients experienced a significant decrease in BMI (44.7 ± 9.5 to 33.8 ± 7.5) and reached 61.2% EWL. A recent multi-center study by Iranmanesh et al. [29], reported longer operative times (203 vs. 154 min, *p* < 0.001), increased number of readmissions for oral intolerance (10.5% vs. 6.7%, *p* = 0.046), and higher rates of gastro-jejunal anastomosis stricture (6.4% vs. 2.7%, *p* = 0.013) during revisional robot-assisted RYGB. However, there were no significant differences in overall and severe complications, anastomotic leak, conversion, or reoperation rates.

Although most studies comparing primary and revisional bariatric surgery are retrospective experiences, they display similar and logical outcomes—increased perioperative complications during revisional cases. Nevertheless, the benefits achieved by RBS in terms of weight loss, resolution of comorbidities, and complications still outweigh the risks of these procedures [30,31]. Yet, adding to the discrepancies in the indications for RBS, there is no clear evidence regarding which surgical approach should be elected.

## 4. Advantages of the Robotic Platform during Revisional Bariatric Surgery

Surgical robots have been designed to excel the limitations of conventional laparoscopy. Whereas laparoscopy involves a two-dimensional view displayed on a monitor, robotic surgery offers a close three-dimensional vision portrayed in the commodity of a console, which gives the surgeon a feeling of operating from inside the cavity. Laparoscopy has been described as counterintuitive, given the mirror-image effect of the camera—when the camera is in front, moving an instrument to the right appears on the left [32]. Moreover, if the camera is unsteadily held by an assistant, the surgeon is forced to adopt unpleasant positions, and the slightest hand tremor is transferred onto the rigid, straight instruments [32]. Robotic systems such as the Da Vinci Xi (Intuitive Surgical, Sunnyvale, CA) provide active camera, multi-quadrant access, improved precision of motion, filtered tremor, and instruments with endowrist movements and seven degrees of freedom, powerfully enhancing the dexterity of the surgeon [32,33]. In addition to the two surgeon arms and the camera commanded from the console, the surgeon can opt to use a third arm to improve traction. Furthermore, the robotic system allows for combination with fluorescence imaging, representing one of the most innovative technologies [34]. Indocyanine green injection produces an angiography-type of image displayed on a monitor, enabling a more precise assessment of bowel perfusion, and guiding decisions on bowel transection intraoperatively.

The robotic platform is advantageous in operations involving reduced, fixed, deep operating fields, or those requiring extreme accuracy, such as micro-anastomosis and fine dissection [35]. This represents the case of RBS, in which adhesions between the internal organs are common, and tissues and vasculature have become frail, producing a hostile abdomen. RBS often implies the confection or re-confection of strenuous intestinal anastomosis, an overwhelming task for laparoscopic instruments. In this scenery, in which an extremely cautious dissection and interpretation of an altered anatomy are crucial, the robot could provide certain benefits.

## 5. Outcomes of Robot-Assisted Revisional Bariatric Surgery

The utilization of the robotic platform for both primary and RBS has experienced a steady increase in the last few years. The robotic approach was first used in the field of bariatric surgery by Himpens et al., who performed a robotic AGB in 1998 [36]. In 2000, Sudan et al. performed and published the first series on robotic biliopancreatic diversion with duodenal switch [37], in which SG was a step of the procedure. Moreover, robot-assisted RYGB was adopted in the early 2000 s with an initially hybrid procedure in which only the hand-sewn gastro-jejunal anastomosis was conducted with the robotic platform [38,39]. Around the year 2008, RYGB was almost fully performed with a robotic system, except for the use of the stapler, which was handled by a bedside assistant, as the first da Vinci stapler was launched in 2014 [40]. The sequential adoption of the robotic platform for primary bariatric procedures, together with the tendency to use this device in more complex cases, resulted in its increasing utilization in RBS in the last decade [41,42]. In effect, Scarritt et al. [42] analyzed the Metabolic and Bariatric Surgery Accreditation and Quality Improvement Program database for the period 2015 to 2018, and found a significant increase in the utilization of the robotic platform for both primary and revisional procedures. The proportion of primary SG, primary RYGB, and revisional cases performed robotically increased from 5.9%, 7.2%, and 1.7% in 2015, to 9.9%, 10.2%, and 3.9% in 2018, respectively [42,43].

Several groups have published their experience with robotic technology in RBS (Table 2). Most of these studies report conversion to RYGB as the most performed robot-assisted RBS. However, the type of primary procedure performed and the participation of other RBS should be scrutinized when analyzing the results. For instance, 42.8% of the patients in the study by Ayloo et al. [44] were conversions from AGB to SG. Similarly, Snyder et al. [45] had 14.1% of these procedures among the subjects, while the RBS analyzed by Cheng et al. [46] included 20.9% redo gastro-jejunostomy. Consistently, AGB was the most frequent index procedure among the studies, and failed (weight-related) primary surgery was the principal indication for revisional surgery.

Concerning operative times, Vilallonga et al. [47] reported the lowest mean value with 180 min, whereas Rebbechi et al. [48] had the highest mean operative time of 265.5 min. Remarkably, the most performed RBS by Vilallonga et al. was the conversion of SG to RYGB, whereas Rebecchi et al. had a higher proportion of conversion from vertical banded gastroplasty to RYGB. As mentioned previously, the variability in the surgical procedures performed could influence these results.

Regarding the complications of robot-assisted RBS, only three studies reported major morbidity outcomes (corresponding to a grade > 2 in the Clavien–Dindo classification) with rates of 2.9%, 3.9%, and 4.5% [46,48,49]. Considering the incidence of postoperative leaks, Dreifuss et al. [49] reported only one case of anastomotic leak after a resection and reconstruction of a SG stenosis, and Cheng et al. [46] reported a patient with abdominal fluid collection after a redo gastro-jejunal anastomosis that was managed with antibiotics and drainage. The conversion, reoperation, and mortality rates were almost negligible among all of the studies.

The mean %EWL, follow-up time, and rates of loss to follow-up were variable among the studies. At 1 year, Rebecchi et al. [48] and Cheng et al. [46] reported 55.4% and 57.6% EWL, respectively, whereas at 2 years, Dreifuss et al. [49] reported a 36.4% EWL and Bindal et al. [50] reported a 60.7 mean %EWL. The study by Bindal et al. [50] focused specifically on weight loss outcomes according to the type of RBS performed, and reported a %EWL at 2 years of up to 51.2% after conversion from SG to RYGB, and up to 70.1% after the conversion of AGB to RYGB.

These results confirm the overall safety of the robotic platform and its satisfactory long-term weight loss outcomes in RBS.

## 6. Robot-Assisted vs. Laparoscopic Revisional Bariatric Surgery

Minimally invasive surgery is the standard of care in bariatric surgery, and the most common procedures can be effectively performed using either conventional laparoscopy or robot-assisted surgery. Regarding this matter, some authors advocate that a robot could have advantageous results in complex procedures [35]. However, results favoring one or the other approach remain ambiguous. Aiming to find certainty, we conducted a literature search for the most recently published studies that compared laparoscopic and robotic approaches in RBS (Table 3).

All of the studies were retrospective. Four out of eight studies used patient data from the Metabolic and Bariatric Surgery Accreditation and Quality Improvement Program. It is important to highlight that a major limitation of this national data registry is the lack of information regarding the index operations, indications for RBS, and for outcomes beyond 30 postoperative days.

The most common index procedure varied among the studies, whereas Gray et al. [51] reported a higher prevalence of AGB (46%), Beckmann et al. [52] had 72.2% prevalence of SG, and King et al. [53] had 40.7% of RYGB. Interestingly, Gray et al. reported significant differences in the indications for RBS, depending on the index procedure: insufficient weight loss represented the main indication for the revision of AGBs (79%), reflux was the main cause in patients who had undergone SG (44%), and anatomic complications constituted 54% of the indications after RYGB.

Conversion to RYGB was the most frequently performed revisional procedure among the studies. In a single-center, retrospective study by Moon et al. [54], statistically significant differences between laparoscopic and robotic groups in the types of RBS were reported. Conversions to complex bariatric procedures (such as RYGB to biliopancreatic diversion with duodenal switch or RYGB to SG) were more common with the robot, whereas the most performed laparoscopic RBS was revision of gastro-jejunostomy. This was presumably related to the surgeons’ preference for the robot-assisted approach for complex procedures.

The operative time was reported in all of the studies but one, significantly favoring the laparoscopic approach. Only the study by Beckmann et al. [52] reported a shorter operative time with the robotic platform (robot-assisted RYGB was completed an average 37 min earlier than laparoscopic RYGB). The authors advocate the rationale for their inconsistency with previous studies in the non-comparability of the other groups in terms of complexity of the revisions performed (i.e., higher proportion of band removals—simpler procedures—in the laparoscopic groups). In addition, the unfavorable operative time of the robot could correspond to the level of experience of the implicated surgeons.

Three studies conducted comparisons of revisional laparoscopic SG vs. robot-assisted SG, and revisional laparoscopic RYGB vs. robot-assisted RYGB [55,56,57]. Acevedo et al. [55] reported a higher rate of transfusion requirements in the laparoscopic RYGB group (2.9% vs. robot-assisted RYGB 0.6%, *p* = 0.02), and a higher rate of postoperative sepsis in the robot-assisted SG group when compared to the laparoscopic group (1.0% vs. 0%, *p* = 0.04). El Chaar et al. [56] only found a trend towards decreased rates of serious adverse events and organ-specific infections in the robot-assisted RYGB group. Interestingly, after adjusting the outcomes for operative time, Nasser et al. [57] still found a higher rate of organ-space surgical site infection and sepsis in robot-assisted SG vs. laparoscopic SG, whereas robot-assisted RYGB had a lower overall morbidity profile (lower rate of respiratory complications, pneumonia, surgical site infection, and bleeding requiring transfusion) than laparoscopic RYGB (AOR 0.74, *p* = 0.01). A significantly higher reoperation rate in patients undergoing revisional robot-assisted SG when compared to laparoscopic SG was also reported, although this difference was no longer seen after adjusting for operative time, a variable that was found to be an independent predictor of morbidity. LOS was inconsistent among the studies, with three studies favoring the laparoscopic approach and three studies favoring the robot. The explanation for these differences remains unclear.

Regarding long-term outcomes, only one study compared weight loss between the groups. A significantly greater change in BMI was evidenced at 30 days in patients undergoing laparoscopic RYGB compared to those undergoing robot-assisted RYGB, and this difference was not appreciated at 1 year of follow-up [52].

An interesting variable only studied by Beckmann et al. [52] is C-reactive protein, a well-known inflammation mediator. C-reactive protein levels at postoperative days 1 and 2 were significantly lower in robot-assisted RYGB patients. Although this outcome could not be associated with complication rates, the authors suggested it could be related to more precise and atraumatic surgery when using the robot [52].

In summary, the reported outcomes of the robotic platform seem ambiguous and heterogeneous when compared to the laparoscopic approach. On this matter, Bertoni et al. [59] recently published a systematic review and meta-analysis, and found similar postoperative complications, conversions, LOS, and operative times between the laparoscopic and robotic approaches. However, LOS and operative time displayed a high heterogeneity, hospital readmissions were significantly higher with the robotic approach (7.1% vs. 5.6%), and included studies were subject to a high risk of selection bias. The authors concluded that robot-assisted revisional bariatric surgery has no significant advantage, however, the robotic approach showed a non-inferior efficacy compared to standard laparoscopy.

Analyzing the outcomes of robot-assisted RBS is complicated due to multiple factors. Firstly, studies reporting perioperative outcomes on robot-assisted RBS and robotic vs. laparoscopic RBS are scarce, retrospective, and mainly have short follow-up periods. Secondly, as previously mentioned, RBS encompasses a wide variety of procedures, so comparing only the approach seems inappropriate. Third, few studies report the details of the index procedure (i.e., approach, surgical technique, and time to revision), and the peculiarities of the revisional procedure performed. In addition, there is heterogeneity in the reporting of complications—while some use the Clavien–Dindo classification, others report specific adverse outcomes or just describe them. In addition, there are practically no definitions of operative time, the robotic expertise of the surgeons implied is not specified, and there is limited information on the long-term outcomes of interest, such as weight loss. All these factors contribute to a general inability to draw major conclusions on the use of the robot for RBS. To do so, randomized controlled trials comparing robotic vs. laparoscopic approaches for specific RBS procedures are required.

## 7. The Learning Curve

One of the most appealing aspects of the robot is its learning curve. Although the number of cases required to reach the learning curve for robot-assisted bariatric surgery has not been established, several studies illustrate a decreased learning curve with the use of this technology when compared to its laparoscopic analogous [60,61,62,63].

On this matter, Vilallonga et al. set 19 robot-assisted SG as the number needed to achieve a decrease in operative time and reach a plateau [60], and recommend this procedure as a good starting point before moving to more complex operations such as robot-assisted RYGB. Zacharoulis et al. set this number in 68 robot-assisted SG, and evidenced that LOS could also significantly decrease with experience [61]. Romero et al. illustrated a tendency towards a reduction in the operative time of robot-assisted SG after the first 25 cases [62]. Buchs et al. [63] published their experience with 64 robot-assisted RYGB, and found significantly decreased operative times after the initial 14 cases.

Regarding the learning curve for laparoscopic operations, Wehrtmann et al. [64] determined that 30–50, 60–100, and 100–200 laparoscopic SG were needed to achieve “competency”, “proficiency”, and “mastery”, respectively, whereas 30–70, 70–150, and up to 500 cases were needed for laparoscopic RYGB, respectively. The number of laparoscopic cases reported in this systematic review seemed significantly higher than those reported for robotic procedures. This provides additional evidence suggesting the robotic platform has a faster learning curve.

Nevertheless, the robotic platform is still an emerging technology, and most surgeons have not met its learning curve yet. Hence, the robotic surgical outcomes reported in the literature may fail to reflect the real advantages of the device.

## 8. Disadvantages of the Robotic Approach

Almost unanimously, the major drawbacks of the robot are the prolonged operative times and higher costs. The up-front cost to purchase the Da Vinci Surgical System is estimated at 1 to over 2 million US dollars, with annual maintenance costs of about 10% of this price, plus instrument expenditures [65]. Beckmann et al. estimated the average cost of the operating room at 15 EUR/minute [52]. In an assessment on the daily expenses of the device, King et al. [66] compared the costs of robot-assisted vs. laparoscopic RYGB, and found no differences in overall costs ($6431.34 vs. $6349.09); instead, the robotic platform had a lower cost of supplies and a trend towards decreased LOS. Factors contributing to differences in costs were presumably related to the instruments used (staplers), technique (hand-sewn anastomosis), surgeon’s experience, and reduced LOS. Certainly, additional variables should be considered when investigating costs. For instance, if we consider that the use of the robot allows for performing an operation with a reduced number of assistants, and salary fares are considered, the cost analysis would probably reveal a profile in favor of the robotic technology [67]. However, systematic reviews on the costs of the robotic approach for bariatric surgery are scarce, and studies lack transparency in their cost-reporting methods, essentially ignoring important components [68].

Another disadvantage of robotic technology is its increased operative time. Operative time depends on the learning curve and is directly related to the costs and complications of the operation. Crucially, the learning curve is not just about the individual technical skills of a surgeon, but also involves the knowledge and capacity of the surgical team during set up, docking, and instrument exchange [60]. As with every new device, it takes time for the surgical team to become acquainted. Finally, once the learning curve of the platform is overcome and better outcomes (i.e., complications) are achieved, the cost-effectiveness of the robot will probably outshine conventional laparoscopy [66].

An infrequently reported, but not minor, disadvantage of the robot is the relative impairment to execute fast changes in patient positioning, as these require removing the instruments and re-docking the platform. This makes the platform less suitable for procedures involving major postural changes during different steps of the operation [35]. In addition, evidence on the role of the robot in surgical emergencies is scarce and remains under investigation [69].

## 9. Conclusions and Future Perspectives

The potential of the robot in RBS seems promising if we consider that this technology has already demonstrated perioperative results comparable to those of conventional laparoscopy. The published studies may be reporting outcomes of an early stage of device-training. As aforementioned, the learning curve is directly related to the increased operative time and costs, so improved outcomes are expected to become evident once this liability is overcome.

## Figures and Tables

**Table 1 jcm-11-01820-t001:** Indications, types, and procedures of RBS according to the most common primary bariatric operations.

Primary Procedure	Indication	Type of RBS	RBS
AGB	Weight regain/inadequate weight loss/comorbidity recurrence	Conversion	- Conversion to SG - Conversion to RYGB - Conversion to BPD/DS
	Complications:		
	Slippage	Correction/Reversal	- Band relocation - Band removal
	Erosion	Reversal	- Band removal
	Intolerance	Correction/Reversal	- Band relocation - Band removal
	Pouch dilation	Reversal	- Band removal
	Port complication	Correction	- Port inspection - Band removal
SG	Weight regain/inadequate weight loss/comorbidity recurrence	Conversion/revision	- Re-sleeve gastrectomy - Conversion to RYGB - Conversion to BPD/DS
	Complications:		
	Stricture	Correction/conversion	- Endoscopic dilation - Re-sleeve gastrectomy - Conversion to RYGB
	Gastroesophageal reflux disease	Correction	- Endoscopic treatment (i.e., Stretta) - Magnetic sphincter augmentation (Linx^®^) - Conversion to RYGB
	Fistula	Revision/conversion	- Reinforcement of the staple line - Endoscopic management - Conversion to RYGB
	Dilation of the reservoir	Correction	- Re-sleeve gastrectomy
RYGB	Weight regain/inadequate weight loss/comorbidity recurrence	Revision/conversion	- Pouch and GJ redo - Conversion to distal RYGB - Conversion to BPD/DS
	Complications:		
	Marginal ulcer	Revision/conversion/reversal	- GJ redo - Reversal - Total gastrectomy (refractory cases)
	Fistula	Correction	- Endoscopic management - Fistulectomy - Gastric remnant resection/trimming - Pouch/GJ redo
	Candy cane syndrome	Correction	- Candy cane resection - GJ redo
	Internal hernia	Correction	- Hernia reduction and closure of mesenteric spaces
	Pouch dilation/stenosis	Correction	- Endoscopic dilation - Pouch trimming - Pouch/GJ redo
	GJ anastomosis dilation/stenosis	Correction	- Endoscopic dilation - GJ redo
	Jejuno−jejunal anastomosis stenosis/stricture	Correction	- Jejuno-jejunal anastomosis redo
	Malabsorption	Reversal	- Reversal

AGB: adjustable gastric banding; SG: sleeve gastrectomy; RYGB: Roux-en-Y gastric bypass; BPD/DS: biliopancreatic diversion with duodenal switch; GJ: gastro-jejunal anastomosis.

**Table 2 jcm-11-01820-t002:** Outcomes of robot-assisted revisional bariatric surgery.

Study	Year Published	n	Main Primary Procedure	Main Indication for Revision	Revisional Procedure Performed	Operative Time (Minutes)	Morbidity	Mortality, n (%)	Reoperation, n (%)	Conversion, n (%)	LOS (Days)	%EWL at End of Follow-Up	Follow Up (Months)
Snyder et al. [45]	2013	99	AGB (65.7%)	Failed AGB 35.4%	Conversion to RYGB 80 (80.8%)	203.8 ± 100 (r: 64–690)	Overall: 17%	0	0	0	2.3 ± 1	60%	36
Ayloo et al. [44]	2015	14	AGB (78.5%)	Weight-related 57.1%	Conversion to SG (42.8%) or RYGB (35.7%)	220.6 ± 64.3	0	0	1 (7%)	0	3.3 ± 1.4	-	6
Bindal et al. [50]	2015	32	AGB (50%)	Weight-related 62.5%	Conversion to RYGB (100%)	226 ± 45.3	-	0	0	0	3 ± 2.6	60.70%	24
Rebecchi et al. [48]	2019	68	Vertical banded gastroplasty (63.2%)	Persistent dysphagia 33.8% Weight-related 33.8%	Conversion to RYGB (100%)	265.6 ± 54.1	Overall: 8.8%. Major: 2.9%	0	1 (1.5%)	2 (2.9%)	5.5 ± 3.9	55.4 ± 34.7%	12
Dreifuss et al. [49]	2020	76	AGB (50%)	Weight-related 76.3%	Conversion to RYGB: 60 (78.9%)	182 (r: 74–376)	Major: 3.9%	1 (1.3%)	3 (3.9%)	0	2.1 (r: 1–18)	36.40%	24.4
Cheng et al. [46]	2021	67	SG (38.8%)	Weight-related 50.7%	Conversion to RYGB: 49 (73.1%)	184.07 ± 54.59	Major: 4.5%	0	2 (3%)	4 (6.0%)	2.46 ± 1.4	57.62%	12
Vilallonga et al. [47]	2021	17	SG (88.2%)	-	Conversion to RYGB (52.9%)	180 (r: 150–240)	Overall: 5.88%	0	1 (5.88%)	-	2.4	-	-

AGB: adjustable gastric banding; SG: sleeve gastrectomy; RYGB: Roux-en-Y gastric bypass. LOS: length of hospital stay. %EWL: percentage of excess weight loss. r: range. Major morbidity: complications corresponding to >2 of the Clavien–Dindo classification.

**Table 3 jcm-11-01820-t003:** Laparoscopic vs. robot-assisted revisional bariatric surgery.

Study	Year Published	n		Operative Time (Minutes)		Morbdity Rate at 30 Days n, (%)		Mortality, n (%)		Reoperation, n (%)		Conversion, n (%)		LOS (Days)		%EWL at End of Follow-Up		Follow Up (Months)
		Lap	R	Lap	R	Lap	R	Lap	R	Lap	R	Lap	R	Lap	R	Lap	R	
Gray et al. [51]	2018	66	18	ABG: 177 ± 71 CSP: 238 ± 81	ABG: 205 ± 101 CSP: 193 ± 41	Overall: AGB: 4 CSP: 10	Overall: AGB: 2 CSP: 2	0	0	-	-	ABG: 0 CSP: 2	ABG: 0 CSP: 0	ABG: 3.7 ± 1.2 CSP: 5.8 ± 3.3 *	ABG: 3.7 ± 1.5 CSP: 3.7 ± 1.7 *	-	-	3
Clapp et al. [58] ⤉	2019	22547	1525	103.7 (SD 67.7) *	167.7 (SD 82.7) *	NS **		52 (0.1%)	4 (0.2%)	-	-	-	-	1.7 (SD 2.8) *	2.3 (SD 3.1) *	-	-	1
Acevedo et al. [55] ⤉	2020	1144	1144	121.7 ± 67.5 *	177.4 ± 79.4 *	NS **		2 (0.2%)	2 (0.2%)	32 (2.8%)	42 (3.7%)	11 (1%)	13 (1.1%)	2.2 ± 3.1 *	2.4 ± 3.1 *	-	-	1
El Chaar et al. [56] ⤉	2020	220	220	127.5 (r: 23–411) *	159 (r: 42–504) *	17 (7.7%)	14 (6.4%)	-	-	8 (3.6%)	6 (2.7%)	-	-	-	-	-	-	1
Nasser et al. [57] ⤉	2020	SG 15935	1077	101.9 ± 48.2 *	145.2 ± 57.4 *	Overall: 4.5% *	Overall: 6.7 *	17 (0.1%)	0	1.5% *	2.4% *	22 (0.1%)	3 (0.3%)	1.7 ± 1.7 *	1.9 ± 2.7 *	-	-	1
		RYGB 11212	1230	153.9 ± 72.0 *	196.7 ± 72.0 *	Overall: 11.6% *	Overall: 9.3% *	21 (0.2%)	1 (0.1%)	3.90%	3.80%	65 (0.6%)	8 (0.7%)	2.4 ± 2.8	2.4 ± 2.5	-	-	
Beckmann et al. [52]	2020	18	41	167.6 ± 33.8 *	130.7 ± 40.4 *	4 (22.2%)	3 (7.3%)	-	-	2 (11.1%)	1 (2.4%)	-	-	6.2 ± 1.6 *	4.9 ± 1.0 *	-	-	12
Moon et al. [54]	2020	64	30	113.3 (SD: 46.2) *	155.5 (SD: 51.2) *	-	-	-	-	0	1 (3.3%)	-	-	2.0 (SD: 1.5)	2.5 (SD: 1.5)	-	-	1
King et al. [53]	2021	115	52	-	-	1 (1.9%)	6 (5.2%)	0	0	-	-	0	0	62.6 h *	40.2 h *	-	-	-

Lap: laparoscopic; R: robot-assisted; AGB: adjustable gastric banding; SG: sleeve gastrectomy; RYGB: Roux-en-Y gastric bypass; CSP: conversion from stapled procedure; LOS: length of hospital stay; %EWL: percentage of excess weight loss. * Statistically significant difference; NS: not significant; SD: standard deviation; r: range. ** No overall or major morbidity rates, but extensive recount of specific individual complications. ⤉ Metabolic and Bariatric Surgery Accreditation and Quality Improvement Program data registry. Major morbidity: complications corresponding to >2 of the Clavien–Dindo classification.

## Data Availability

Not applicable.

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
