# Peer review of "Current Status of Robot-Assisted Revisional Bariatric Surgery"

_jcm, 2022, doi:10.3390/jcm11071820_

Round 1
Reviewer 1 Report
In this narrative review, Dr. Vanetta and colleagues discussed the promising future of robotic surgery for revisional bariatric surgery. They concluded that this approach is safe and feasible, although long surgery duration and high cost remain the major drawbacks. Overall, the manuscript was written nicely and it has clinical insights. Nonetheless, some information are missing and need to be added:
- "the prevalence of obesity has reached 41.9% among adults, and 19.7% among children and adolescents aged 2-19 years old" I think, in addition to this sentence, the authors need to add the epidemiology of morbid obesity and the proportion of them who need bariatric surgery. This is important because not all obesity needs BS, majority of them can be treated using lifestyle modification and pharmacological treatments.
- Line 29: "...well-known comorbidities..." please specify these comorbidities.
- "bariatric surgery has demonstrated to be effective in achieving significant weight loss and remission of obesity-related comorbidities." Please add the fact that BS also reduces MACE, as previously shown in this meta-analysis (PMID: 34684569).
- I think it is also important to briefly explain the superiority of BS as compared with lifestyle modifications in short and long term.
- "Accounting for 6.0% of the bariatric procedures in 2011, they peaked to 16.7% in 2019 in the US" Please explain the reasons. I can understand if the absolute number was increased because the number of BS increased over time but the increased frequency of BS does not explain the (almost) tripling of the percentage of revisional BS.
- Line 50: please remove "Logically, "
- Please remove section "Materials and Methods" because this is a narrative review. If the authors aimed to write a systematic review, it has to follow PRISMA guideline, the authors have to expand the search databases, keywords, and the authors need to register it in PROSPERO.
- Line 264: What was the basis of saying "In summary, the robotic platform has proven at least its non-inferiority when compared to conventional laparoscopy in RBS"? This statement was not reflected from the data in Table 3. In that table, RBS was associated with longer surgical time and a significantly higher reoperation rate. Please clarify this and perhaps if the results of studies are inconclusive, the authors could use other words than "has proven", which indicates full certainty.
- Could the authors check whether this data has been included (PMID: 34248332)? They have a subset of patients underwent robotic revisional BS as well.
Author Response
Reviewer #1
In this narrative review, Dr. Vanetta and colleagues discussed the promising future of robotic surgery for revisional bariatric surgery. They concluded that this approach is safe and feasible, although long surgery duration and high cost remain the major drawbacks. Overall, the manuscript was written nicely and it has clinical insights. Nonetheless, some information are missing and need to be added:
- Comment 1.1: "the prevalence of obesity has reached 41.9% among adults, and 19.7% among children and adolescents aged 2-19 years old" I think, in addition to this sentence, the authors need to add the epidemiology of morbid obesity and the proportion of them who need bariatric surgery. This is important because not all obesity needs BS, majority of them can be treated using lifestyle modification and pharmacological treatments.
Response 1.1: Thank you very much for reviewing our manuscript and for your comments. We agree that adding the epidemiology of obesity and the proportion of patients requiring bariatric surgery is important.
The following information has been added to the introduction section of the manuscript:
“Specifically, severe obesity, corresponding to a BMI of 40 or higher, was present in 9.2% of U.S. adults [https://asmbs.org/resources/obesity-in-america]”. “Despite these alarming figures, the American Society for Metabolic and Bariatric Surgery (ASMBS) estimates indicate that less than 1% of the population eligible for bariatric surgery actually received it in 2019 [https://asmbs.org/resources/obesity-in-america], a dissociation which is believed to correspond to an unequal access to surgery, related to income and education level, among other factors [https://bariatrictimes.com/equity-in-bariatric-surgery-march-2018].”
- Comment 1.2: Line 29: "...well-known comorbidities..." please specify these comorbidities.
Response 1.2: Obesity-related comorbidities have been specified as suggested.
The following sentence of the introduction section of the manuscript has been modified:
“The relationship between obesity and the development of various well-known comorbidities, including cardiovascular diseases, diabetes, liver disease, chronic kidney disease, osteoarthritis and cancer, positions this condition as one of the top preventable causes of mortality.”
- Comment 1.3: "bariatric surgery has demonstrated to be effective in achieving significant weight loss and remission of obesity-related comorbidities." Please add the fact that BS also reduces MACE, as previously shown in this meta-analysis (PMID: 34684569).
Response 1.3: Thanks for raising this interesting point. The information was added to the introduction section of the manuscript:
“On the contrary, bariatric surgery has demonstrated to be effective in achieving long-term weight loss (up to 77% of excess weight at 1 year and more than 50% at 10-20 years [https://asmbs.org/resources/obesity-in-america][PMID: 30293134]), remission of obesity-related comorbidities, and reducing the incidence of major cardiovascular events [PMID: 34684569].”
- Comment 1.4: I think it is also important to briefly explain the superiority of BS as compared with lifestyle modifications in short and long term.
Response 1.4: Thanks for your suggestion. We have included the following information addressing the superiority of BS compared to lifestyle modifications:
“On this matter, intense lifestyle modification and pharmacotherapy usually fail to achieve long-term sustained weight loss and remission of obesity-related comorbidities (10% weight loss at 1 year and 5.3% at 8 years) [PMID: 27230645]. On the contrary, bariatric surgery has demonstrated to be effective in achieving long-term weight loss (up to 77% of excess weight at 1 year and more than 50% at 10-20 years [https://asmbs.org/resources/obesity-in-america][PMID: 30293134]), remission of obesity-related comorbidities, and reducing the incidence of major cardiovascular events [PMID: 34684569].”
- Comment 1.5: "Accounting for 6.0% of the bariatric procedures in 2011, they peaked to 16.7% in 2019 in the US" Please explain the reasons. I can understand if the absolute number was increased because the number of BS increased over time but the increased frequency of BS does not explain the (almost) tripling of the percentage of revisional BS.
Response 1.5: This is a very interesting observation that caught our attention too. It is hard to say exactly why this phenomenon is occurring. We believe that the increased frequency of revisional cases is multifactorial: increased rates of obesity and primary bariatric procedures, changes in the operations performed over time, failure of some procedures to achieve sustained weight loss, increase in the available literature supporting the benefits of revisional bariatric surgery, and surgeon’s experience with revisional cases.
The following comment was added to the introduction section of the manuscript:
“Although the trending increase in the percentage of RBS relative to other bariatric procedures is difficult to explain, it could be related to multiple factors such as higher rates of obesity and primary bariatric surgeries, a shift in the types of bariatric procedures being performed, failure of some procedures to achieve sustained weight loss, increase in the available literature supporting the benefits of revisional bariatric surgery, and an increased experience with revisional cases.”
- Comment 1.6: Line 50: please remove "Logically, "
Response 1.6: “Logically” was removed.
- Comment 1.7: Please remove section "Materials and Methods" because this is a narrative review. If the authors aimed to write a systematic review, it has to follow PRISMA guideline, the authors have to expand the search databases, keywords, and the authors need to register it in PROSPERO.
Response 1.7: Thank you very much for this comment. We agree that our manuscript is a narrative review, and the section was removed as suggested.
- Comment 1.8: Line 264: What was the basis of saying "In summary, the robotic platform has proven at least its non-inferiority when compared to conventional laparoscopy in RBS"? This statement was not reflected from the data in Table 3. In that table, RBS was associated with longer surgical time and a significantly higher reoperation rate. Please clarify this and perhaps if the results of studies are inconclusive, the authors could use other words than "has proven", which indicates full certainty.
Response 1.8: We agree that this sentence might be misleading. This statement was referring to the systematic review by Bertoni et al., who conducted a meta-analysis on robotic vs laparoscopic approach for revisional bariatric surgery. We have rephrased the paragraph and included relevant conclusions of this study:
“In summary, the reported outcomes of the robotic platform seem ambiguous and heterogeneous when compared to the laparoscopic approach. On this matter, Bertoni et al. [​​PMID: 34410582] recently published a systematic review and meta-analysis, and found similar postoperative complications, conversions, LOS, and operative times between the laparoscopic and robotic approaches. However, LOS and operative time displayed high heterogeneity, hospital readmissions were significantly higher with the robotic approach (7.1% vs 5.6%), and included studies were subject to a high risk of selection bias. The authors concluded that robotic-assisted revisional bariatric surgery has no significant advantage, however, the robotic approach showed a non-inferior efficacy compared to standard laparoscopy.”
- Comment 1.9: Could the authors check whether this data has been included (PMID: 34248332)? They have a subset of patients who underwent robotic revisional BS as well.
Response 1.9: Thank you very much for the suggestion. We have included some relevant data of this article on the section entitled “Revisional bariatric surgery: a challenging operation”:
“Revisional surgery implies operating on the same structure for a second time. Hence, increased perioperative complications should be expected. On this matter, Howell et al. [PMID: 34248332] reported significantly higher morbidity rates for revisional vs primary bariatric surgery (14.8% vs 3.9%). [...]”
Reviewer 2 Report
The review by Vanetta et al titled “Current status of robotic-assisted revisional bariatric surgery” aims to examine the most current and representative literature on the outcomes of robotic-assisted revisional bariatric surgery (RBS). Although the article was well written and is interesting to read, there are minor comments that need to be looked at before being considered for publication. The formatting of the article is not standard.
The results and writing in general did not look into the outcomes of the robotic-assisted RBS but revisional surgeries in general. In this instance it will be better to give a suitable title to the article, rather than the one provided.
The methodology section for the selection of the articles for the review needs to be described in detail. Additionally both the results and discussion sections are missing from the article and it’s not clear where the methods section ended and the other section started. The article lacks details on the robotic procedures that have been used, either historically or recently. Adding this will add to the review.
Author Response
Reviewer #2
The review by Vanetta et al titled “Current status of robotic-assisted revisional bariatric surgery” aims to examine the most current and representative literature on the outcomes of robotic-assisted revisional bariatric surgery (RBS). Although the article was well written and is interesting to read, there are minor comments that need to be looked at before being considered for publication.
- Comment 2.1: The formatting of the article is not standard.
Response 2.1: Thank you very much for taking the time to review our work and for your comments. We are glad you found our manuscript well-written and interesting. This manuscript represents a narrative review and was written following the Journal of Clinical Medicine guidelines. The review is not systematic, and therefore, it is not following a structured format. However, we decided to arrange it in this way since we thought it results in an appealing reader-friendly format.
- Comment 2.2: The results and writing in general did not look into the outcomes of the robotic-assisted RBS but revisional surgeries in general. In this instance it will be better to give a suitable title to the article, rather than the one provided.
Response 2.2: Thank you very much for your comment. We believe it is relevant to introduce the obesity pandemic and talk about revisional bariatric surgery in general (indications, higher complexity, etc.) before expanding on robotic revisional surgery. We certainly included a lot of information on the text and tables regarding the outcomes of robotic revisional bariatric surgery, advantages and disadvantages of the robotic approach, learning curve, and outcomes comparisons with standard laparoscopy.
If reviewers/editors agree, an alternative title could be “Current status of minimally invasive revisional bariatric surgery”. We hope you find it more suitable for our manuscript.
- Comment 2.3: The methodology section for the selection of the articles for the review needs to be described in detail. Additionally both the results and discussion sections are missing from the article and it’s not clear where the methods section ended and the other section started.
Response 2.3: Thanks for your comment. Following the Journal of Clinical Medicine guidelines for Reviews and per Reviewer #1 request, we have erased the Methods section of our manuscript. This was done because our manuscript is a narrative and not a systematic review. On this matter, JCM allows for narrative reviews to follow a free format, as opposed to systematic reviews in which the standardized introduction, material and methods, results, and discussion sections are required.
- Comment 2.4: The article lacks details on the robotic procedures that have been used, either historically or recently. Adding this will add to the review.
Response 2.4: Following your suggestion, we further expanded the section on “Outcomes of robotic revisional bariatric surgery”. The following paragraph on the history of the robotic platform in bariatric surgery and its adoption in revisional bariatric surgery has been added:
“The utilization of the robotic platform for both primary and revisional bariatric surgery has experienced a steady increase in the last few years. The robotic approach was first used in the field of bariatric surgery by Himpens et al. who performed a robotic AGB in 1998 [PMID: 10340781]. In 2000, Sudan et al. performed and published the first series on robotic BPD/DS [PMID: 17308948], in which SG was a step of the procedure. Moreover, robotic RYGB was adopted in the early 2000s with an initially hybrid procedure in which only the hand-sewn gastrojejunal anastomosis was conducted with the robotic platform [PMID: 18071805][PMID: 17063303]. Around the year 2008, robotic RYGB was almost fully performed with the robotic system, except for the use of the stapler which was handled by a bedside assistant, since the first da Vinci stapler was launched in 2014 [PMID: 16756726]. The sequential adoption of the robotic platform for primary bariatric procedures, together with the tendency to use this device in more complex cases, resulted in its increasing utilization in RBS in the last decade [PMID: 23778837][PMID: 33165753]. In effect, Scarritt et al. [PMID: 33165753] analyzed the MBSAQIP database for the period 2015 to 2018 and found a significant increase in the utilization of the robotic platform for both primary and revisional procedures. The proportion of primary SG, primary RYGB, and revisional cases performed robotically increased from 5.9%, 7,2%, and 1.7% in 2015, to 9.9%, 10.2%, and 3.9% in 2018, respectively [PMID: 33165753][PMID: 33783678].”
We also further specified the types of robotic RBS performed in the studies. We agree that this information is highly valuable and adds to the review and to better understand the outcomes:
“However, the type of primary procedure performed and the participation of other RBS should be scrutinized when analyzing results. For instance, 42.8% of the patients in the study by Ayloo et al. [PMID: 25303498] were conversions from AGB to SG. Similarly, Snyder et al. [PMID: 23456226] had 14.1% of these procedures among the subjects, while the RBS analyzed by Cheng et al. [PMID: 33646519] included 20.9% redo gastrojejunostomy.”
Reviewer 3 Report
Congratulations to the authors for this comprehensive, well written manuscript.
There are a few minor points that I would like to address:
- ll. 45-46 "Estimates indicate that the global incidence of RBS is 5-54%" : the range of 5-54% is somewhat wide and the cited reference a bit outdated. please try to specify here.
- ll. 84-85: consider citing the systematic review by Mann et al. (PMID 25515500) for the most common definition of <50% EWL here; there is also a time point of 18 months indicated
- Table 2: Try to compact the table to simplify reading; i.e. where is the difference between "Inadequate weight loss", "Weight loss failure"?
- Is there any information on the percentage of centers/surgeons actually using the robotic approach?
Author Response
Reviewer #3
Congratulations to the authors for this comprehensive, well written manuscript. There are a few minor points that I would like to address:
- Comment 3.1: ll. 45-46 "Estimates indicate that the global incidence of RBS is 5-54%" : the range of 5-54% is somewhat wide and the cited reference a bit outdated. please try to specify here.
Response 3.1: First of all, we are glad you found our manuscript comprehensive and well written, and we would like to thank you for taking the time to review our work.
We agree with your comment, this range is indeed wide. We have updated this information with a more recent source published in SOARD in 2018:
“Recent estimates indicate that the incidence of RBS ranges from less than 5% to 26% [PMID: 29496440]”
- Comment 3.2: ll. 84-85: consider citing the systematic review by Mann et al. (PMID 25515500) for the most common definition of <50% EWL here; there is also a time point of 18 months indicated
Response 3.2: Thank you very much for this valuable suggestion. We have cited the review by Mann and rearranged the paragraph as follows:
“Unfortunately, there is no consensus in the definition of “failure” of bariatric surgery. Weight-related failure typically involves either inadequate weight loss or weight regain. The systematic review by Mann et al. [PMID: 25515500] found that the most frequently used definition of inadequate weight loss was <50% excess weight loss (EWL) at 18 months. The second most frequent definition was <25% EWL without a specific time-frame. In addition, weight regain has been defined as progressive weight increase after the achievement of an initially successful weight loss or nadir weight [PMID: 33555451]. However, some authors have argued that EWL is highly dependent on the preoperative BMI and fails to reflect the real success. Instead, they propose the percentage of the total weight (%TWL), with a lower limit of 20%, as the best indicator of success [PMID: 31256357][PMID: 23434275]. In any case, the majority of studies fail to report the parameters and values used to define failure [PMID: 25515500].”
- Comment 3.3: Table 2: Try to compact the table to simplify reading; i.e. where is the difference between "Inadequate weight loss", "Weight loss failure"?
Response 3.3: This is a very good suggestion. We have simplified this parameter in our Table. We switched to “Weight-related” as the preferred term to describe patients that underwent revisions due to weight loss failure, inadequate weight loss, or weight regain.
- Comment 3.4: Is there any information on the percentage of centers/surgeons actually using the robotic approach?
Response 3.4: Thanks for raising this interesting point. The robotic approach for bariatric surgery is gaining popularity. Unfortunately, we did not find any data regarding the percentage of surgeons/centers using the robotic approach. However, two recently published MBSAQIP database analyses showed a significant increase in the utilization of the robotic platform for both primary and revisional bariatric procedures. The following information was added to the “Outcomes of robotic revisional bariatric surgery” section of the manuscript:
“In effect, Scarritt et al. [PMID: 33165753] analyzed the MBSAQIP database for the period 2015 to 2018 and found a significant increase in the utilization of the robotic platform for both primary and revisional procedures. The proportion of primary SG, primary RYGB, and revisional cases performed robotically increased from 5.9%, 7,2%, and 1.7% in 2015, to 9.9%, 10.2%, and 3.9% in 2018, respectively [PMID: 33165753][PMID: 33783678].”
Round 2
Reviewer 1 Report
Thanks for the responses. I have no further comment.
Author Response
Thank you for your time, and for helping us improve our manuscript with your comments and suggestions.